# Chain-of-Factors Paper-Reviewer Matching

## ABSTRACT

With the rapid increase in paper submissions to academic conferences, the need for automated and accurate paper-reviewer matching is more critical than ever. Previous efforts in this area have considered various factors to assess the relevance of a reviewer's expertise to a paper, such as the semantic similarity, shared topics, and citation connections between the paper and the reviewer's previous works. However, most of these studies focus on only one factor, resulting in an incomplete evaluation of the paper-reviewer relevance. To address this issue, we propose a unified model for paper-reviewer matching that jointly considers semantic, topic, and citation factors. To be specific, during training, we instruction-tune a contextualized language model shared across all factors to capture their commonalities and characteristics; during inference, we chain the three factors to enable step-by-step, coarse-to-fine search for qualified reviewers given a submission. Experiments on four datasets (one of which is newly contributed by us) spanning various fields such as machine learning, computer vision, information retrieval, and data mining consistently demonstrate the effectiveness of our proposed CHAIN-OF-FACTORS model in comparison with state-of-the-art paper-reviewer matching methods and scientific pre-trained language models.

## CCS CONCEPTS

• **Information systems** → **Retrieval models and ranking**; • **Computing methodologies** → **Natural language processing**.

## KEYWORDS

paper-reviewer matching; scientific text mining; instruction tuning

**ACM Reference Format:**

Anonymous Author(s). 2024. Chain-of-Factors Paper-Reviewer Matching. In *Proceedings of ACM Conference (Conference'17)*. ACM, New York, NY, USA, 10 pages. https://doi.org/10.1145/nnnnnnn.nnnnnnn

## 1 INTRODUCTION

Finding experts with certain knowledge in online communities has wide applications on the Web, such as community question answering on Stack Overflow and Quora [15, 42] as well as authoritative scholar mining from DBLP and AMiner [12, 61]. In the academic domain, automatic paper-reviewer matching has become an increasingly crucial task recently due to explosive growth in the number of submissions to conferences and journals. Given a huge volume of (e.g., several thousand) submissions, it is prohibitively time-consuming for chairs or editors to manually assign papers

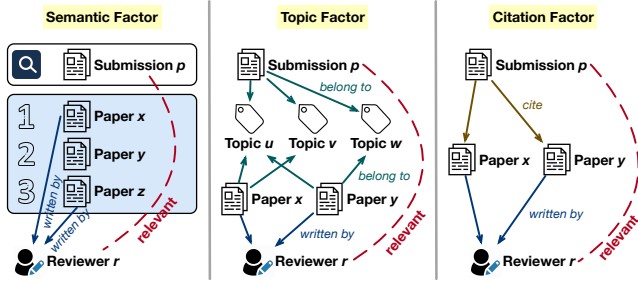

**Figure 1: Three major factors (i.e., semantic, topic, and citation) that should be considered for paper-reviewer matching.**

to appropriate reviewers. Even if reviewers can self-report their expertise on certain papers through a bidding process, they can hardly scan all submissions, hence an accurate pre-ranking result should be delivered to them so that they just need to check a shortlist of papers. In other words, a precise scoring system that can automatically judge the expertise relevance between each paper and each reviewer becomes an increasingly urgent need for finding qualified reviewers.

Paper-reviewer matching has been extensively studied as a text mining task [1, 7, 18, 33, 47, 50], which aims to estimate to what extent a reviewer is qualified to review a submission given the text (e.g., title and abstract) of the submission as well as the papers previously written by the reviewer. Intuitively, as shown in Figure 1, there are three major factors considered by related studies. (1) *Semantic*: Taking the submission $p$ as a query, if the papers most semantically relevant to the query are written by a reviewer $r$, then $r$ should be qualified to review $p$. This intuition is used by previous methods such as the Toronto Paper Matching System (TPMS) [7], where tf–idf is used for calculating the semantic relevance. (2) *Topic*: If a reviewer $r$'s previous papers share many fine-grained research topics with the submission $p$, then $r$ is assumed to be an expert reviewer of $p$. This assumption is utilized by topic modeling approaches [1, 18, 33]. (3) *Citation*: Authors of the papers cited by the submission $p$ are more likely to be expert reviewers of $p$. This intuition is leveraged by studies [47, 50] using citation-enhanced scientific pre-trained language models (PLMs) such as SPECTER [9] and SciNCL [36]. Note that the majority of previous studies do not assume that the topics and references of each paper are provided as input. Instead, such information should be inferred from the paper text.[1]

Although various factors have been explored by previous studies, we find that each method takes only one factor into account in most cases. Intuitively, the semantic, topic, and citation factors correlate with each other but cannot fully replace each other. Therefore,

---

[1] The reasons why related studies make such an assumption are multifold in our view. To be specific, topics selected by the authors when they submit the paper are too coarse (e.g., "Text Mining"), while paper-reviewer matching relies heavily on more fine-grained topics (e.g., "Community Question Answering"); references in the submission do not necessarily cover all papers that ought to be cited, so we should infer what the submission should cite rather than what it actually cites.

considering any of the three factors alone will lead to an incomprehensive evaluation of the paper-reviewer relevance. Moreover, these factors are mutually beneficial. For example, understanding the intent of one paper citing the other helps the estimation of their semantic and topic relevance as well. Hence, one can expect that a model jointly learning these three factors will achieve better accuracy in each factor. Furthermore, the three factors should be considered in a step-by-step, coarse-to-fine manner. To be specific, semantic relevance serves as the coarsest signal to filter totally irrelevant reviewers; after examining the semantic factor, we can classify each submission and each relevant reviewer to a fine-grained topic space and check if they share common fields-of-study; after confirming that a submission and a reviewer's previous paper have common research themes, the citation link between them will become an even stronger signal, indicating that the two papers may focus on the same task or datasets and implying the reviewer's expertise on this submission.

**Contributions.** Inspired by the discussion above, in this paper, we propose a CHAIN-OF-FACTORS framework (abbreviated to CoF) to unify the semantic, topic, and citation factors into one model for paper-reviewer matching. By "unify", we mean: (1) pre-training one model that jointly considers the three factors so as to improve the performance in *each* factor and (2) chaining the three factors during inference to facilitate step-by-step, coarse-to-fine search for expert reviewers. To implement this goal, we collect pre-training data of different factors from multiple sources [9, 47, 63] to train a PLM-based paper encoder. This encoder is shared across all factors to learn common knowledge. Meanwhile, being aware of the uniqueness of each factor and the success of instruction tuning in multitask pre-training [2, 45, 53, 54], we introduce factor-specific instructions to guide the encoding process so as to obtain factor-aware paper representations. Inspired by the effectiveness of Chain-of-Thought prompting [55], given the pre-trained instruction-guided encoder, we utilize semantic, topic, and citation-related instructions in a chain manner to progressively filter irrelevant reviewers.

We conduct experiments on four datasets covering different fields including machine learning, computer vision, information retrieval, and data mining. Three of the datasets are released in previous studies [21, 33, 47]. The fourth is newly annotated by us, which is larger than the previous three and contains more recent papers. Experimental results show that our proposed CoF model consistently outperforms state-of-the-art paper-reviewer matching approaches and scientific PLM baselines on all four datasets. Further ablation studies validate the reasons why CoF is effective: (1) CoF jointly considers three factors rather than just one, (2) CoF chains three factors to facilitate step-by-step, coarse-to-fine relevant paper selection instead of merging all factors in one step, and (3) CoF improves upon the baselines in *each* factor empirically.

## 2 PRELIMINARIES

### 2.1 Problem Definition

Given a set of paper submissions $\mathcal{P} = \{p_1, p_2, ..., p_M\}$ and a set of candidate reviewers $\mathcal{R} = \{r_1, r_2, ..., r_N\}$, the paper-reviewer matching task aims to learn a function $f : \mathcal{P} \times \mathcal{R} \rightarrow \mathbb{R}$, where $f(p, r)$ reflects the expertise relevance between the paper $p$ and the reviewer $r$ (i.e., how knowledgeable that $r$ is to review $p$). We conform to the following three key assumptions made by previous studies

[1, 21, 29, 33, 47]: (1) We do not know any $f(p, r)$ ($p \in \mathcal{P}, r \in \mathcal{R}$) as supervision, which is a natural assumption for a fully automated paper-reviewer matching system. In other words, $f$ should be derived in a zero-shot setting, possibly by learning from available data for other tasks. (2) To characterize each paper $p \in \mathcal{P}$, its text information (e.g., title and abstract) is available, denoted by Text($p$). (3) To characterize each reviewer $r \in \mathcal{R}$, its previous papers are given, denoted by $Q_r = \{q_{r,1}, q_{r,2}, ..., q_{r,|Q_r|}\}$. The text information of each previous paper $q \in Q_r$ is also provided. $Q_r$ is called the *publication profile* of $r$ [34]. In practice, $Q_r$ may be a subset of $r$'s previous papers (e.g., those published within the last 10 years or those published in top-tier venues only). To summarize, the task is defined as follows:

*Definition 2.1.* (Problem Definition) Given a set of papers $\mathcal{P}$ and a set of candidate reviewers $\mathcal{R}$, where each paper $p \in \mathcal{P}$ has its text information Text($p$) and each reviewer $r \in \mathcal{R}$ has its publication profile $Q_r$ (as well as Text($q$), $\forall q \in Q_r$), the paper-reviewer matching task aims to learn a relevance function $f : \mathcal{P} \times \mathcal{R} \rightarrow \mathbb{R}$ and rank the candidate reviewers for each paper according to $f(p, r)$.

After $f(p, r)$ is learned, there is another important line of work focusing on assigning reviewers to each paper according to $f(p, r)$ under certain constraints (e.g., the maximum number of papers each reviewer can review, the minimum number of reviews each paper should receive, and fairness in the assignment), which is cast as a combinatorial optimization problem [17, 20, 23, 24, 30, 38, 49, 56]. This problem is usually studied independently from how to learn $f(p, r)$ [1, 21, 29, 33, 47]. Therefore, in this paper, we concentrate on learning a more accurate relevance function and do not touch the assignment problem.

### 2.2 Semantic, Topic, and Citation Factors

Before introducing our CoF framework, we first examine the factors considered by previous studies on paper-reviewer matching.

**Semantic Factor.** The Toronto Paper Matching System (TPMS) [7] uses a bag-of-words vector (with tf–idf weighting) to represent each submission paper or reviewer, where a reviewer $r$'s text is the concatenation of its previous papers (i.e., $\|_{q \in Q_r}$Text($q$)). Given a paper and a reviewer, their relevance $f(p, r)$ is the dot product of their corresponding vectors. For the perspective of the vector space model [44], each paper $p$ is treated as a "query"; each reviewer $r$ is viewed as a "document"; the expertise relevance between $p$ and $r$ is determined by the similarity between the "query" and the "document", which is the typical setting of semantic retrieval.

**Topic Factor.** Topic modeling approaches such as Author-Persona-Topic Model [33] and Common Topic Model [1] project papers and reviewers into a field-of-study space, where each paper or reviewer is represented by a vector of its research fields. For example, a paper may be 40% about "Large Language Models", 40% about "Question Answering", 20% about "Precision Health", and 0% about other fields. If a paper and a reviewer share common research fields, then the reviewer is expected to have sufficient expertise to review the paper. Intuitively, the field-of-study space needs to be fine-grained enough because sharing coarse topics only (e.g., "Natural Language Processing" or "Data Mining") is not enough to indicate the paper-reviewer expertise relevance.

**Citation Factor.** Recent studies [47, 50] adopt scientific PLMs, such as SPECTER [9] and SciNCL [36], for paper-reviewer matching. During their pre-training, both SPECTER and SciNCL are initialized from SciBERT [6] and trained on a large number of citation links between papers. Empirical results show that emphasizing such citation information significantly boosts their performance in comparison with SciBERT. The motivation of considering the citation factor in paper-reviewer matching is also clear: if a paper $p$ cites many papers written by a reviewer $r$, then $r$ is more likely to be a qualified reviewer of $p$.

Although the three factors are correlated with each other (e.g., if one paper cites the other, then they may also share similar topics), they are obviously not identical. However, most previous studies only consider one of the three factors, resulting in an incomprehensive evaluation of paper-reviewer relevance. Moreover, the techniques used are quite heterogeneous when considering different factors. For example, citation-based approaches [47, 50] already exploit contextualized language models, whereas semantic/topic-based models [1, 7, 33] still adopt bag-of-words representations or context-free embeddings. To bridge this gap, in this paper, we aim to propose a unified framework to jointly consider the three factors for paper-reviewer matching.

## 3 MODEL

### 3.1 Chain-of-Factors Matching

To consider different factors with a unified model, we exploit the idea of instruction tuning [2, 37, 45, 53, 54] and prepend factor-related instructions to each paper to get its factor-aware representations. To be specific, when we consider the **semantic factor**, we can utilize a language model to jointly encode the instruction "Retrieve a scientific paper that is relevant to the query." and a paper $p$'s text to get $p$'s semantic-aware embedding; when we consider the **topic factor**, the instruction can be changed to "Find a pair of papers that one paper shares similar scientific topic classes with the other paper." so that the PLM will output a topic-aware embedding of $p$; when we consider the **citation factor**, we can use "Retrieve a scientific paper that is cited by the query." as the instruction context when encoding $p$. To summarize, given a paper $p$ and a factor $\phi$, where $\phi \in \{\text{semantic}, \text{topic}, \text{citation}\}$, we can leverage a language model to jointly encode a factor-aware instruction $i_\phi$ and $\text{Text}(p)$ to get its $\phi$-aware embedding $g(p|\phi)$.

The detailed architecture and pre-training process of the encoder $g(\cdot|\cdot)$ will be explained in Sections 3.2 and 3.3, respectively. Here, we first introduce how to use such an encoder to perform chain-of-factors paper-reviewer matching, which is illustrated in Figure 2. Given a paper $p$, we let the model select expert reviewers step by step. First, we retrieve papers that are relevant to $p$ from the publication profile of all candidate reviewers. In this step, the semantic factor is considered. Formally,

$$f_{\text{semantic}}(p, q) = g(p|\text{semantic})^\top g(q|\text{semantic}), \quad \forall q \in \cup_{r \in \mathcal{R}} Q_r. \quad (1)$$

Then, we rank all papers in $\cup_{r \in \mathcal{R}} Q_r$ according to $f_{\text{semantic}}(p, \cdot)$ and only select those top-ranked ones (e.g., top 1%) for the next step. We denote the set of retrieved relevant papers as $Q_\mathbb{S}$, where $\mathbb{S}$ stands for the semantic factor.

After examining the semantic factor, we proceed to the topic factor. Intuitively, if a reviewer $r$'s previous papers share fine-grained themes with a submission $p$, we should get a stronger hint of $r$'s

expertise on $p$. Therefore, we further utilize a topic-related instruction to calculate the topic-aware relevance between $p$ and each retrieved relevant paper $q$.

$$f_{\text{topic}}(p, q) = g(p|\text{topic})^\top g(q|\text{topic}), \quad \forall q \in Q_\mathbb{S}. \quad (2)$$

We then rank all papers in $Q_\mathbb{S}$ according to $f_{\text{topic}}(p, \cdot)$ and pick those top-ranked ones as the output of this step, which we denote as $Q_{\mathbb{S} \to \mathbb{T}}$, where $\mathbb{S}$ stands for the semantic factor.

After checking the topic factor, we further consider citation signals. Given that two papers share common fine-grained research topics, the citation link should provide an even stronger signal of the relevance between two papers. For instance, if two papers are both about "Information Extraction", then one citing the other may further imply that they are studying the same task or using the same dataset. However, without the premise that two papers have common research fields, the citation link becomes a weaker indicator. For example, a paper about "Information Extraction" can cite a paper about "Large Language Models" simply because the former paper uses the large language model released in the latter one. This highlights our motivation to chain the three factors for step-by-step, coarse-to-fine selection of relevant papers and expert reviewers. Formally, given $Q_{\mathbb{S} \to \mathbb{T}}$, we use a citation-related instruction to calculate the citation-aware relevance between $p$ and each selected paper.

$$f_{\text{citation}}(p, q) = g(p|\text{citation})^\top g(q|\text{citation}), \quad \forall q \in Q_{\mathbb{S} \to \mathbb{T}}. \quad (3)$$

Finally, we aggregate the score of papers to the score of candidate reviewers writing these papers.

$$f(p, r) = \sum_{q \in Q_r \cap Q_{\mathbb{S} \to \mathbb{T}}} \Big( f_{\text{semantic}}(p, q) + f_{\text{topic}}(p, q) + f_{\text{citation}}(p, q) \Big). \quad (4)$$

Here, $f(p, r)$ is the final relevance score between $p$ and $r$, which can be used to rank all candidate reviewers for $p$. Note that in the last step, we consider the sum of three types of relevance, so our chain-of-factors matching strategy can be denoted as $\mathbb{S} \to \mathbb{T} \to \mathbb{S} + \mathbb{T} + \mathbb{C}$. In our experiments (Section 4.3), we will demonstrate its advantage over only considering the citation factor in the last step (i.e., $\mathbb{S} \to \mathbb{T} \to \mathbb{C}$).

### 3.2 Instruction-Guided Paper Encoding

Now we introduce the details of our proposed encoder $g(\cdot|\cdot)$ that can jointly encode a factor-aware instruction and a paper's text information. This section will focus on the architecture of this encoder, and Sections 3.3 will elaborate more on its pre-training process.

In CoF, we propose to pre-train two text encoders, one for encoding instructions and the other for encoding papers given instruction representations as contexts.

**Instruction Encoding.** Given an instruction $i_\phi$ (which is a sequence of tokens $z_1 z_2 ... z_A$), the instruction encoder $\text{Enc}_i(\cdot)$ adopts a 12-layer Transformer architecture [52] (i.e., the same as $\text{BERT}_{\text{base}}$ [13]) to encode $i_\phi$. Formally, let $\boldsymbol{h}_z^{(0)}$ denote the input representation of token $z$ (which is the sum of $z$'s token embedding, segment embedding, and position embedding according to [13]); let $\boldsymbol{h}_z^{(l)}$ denote the output representation of $z$ after the $l$-th layer. Then, the entire instruction $i_\phi$ can be represented as $H_{i_\phi}^{(l)} = [\boldsymbol{h}_{z_1}^{(l)}, \boldsymbol{h}_{z_2}^{(l)}, ..., \boldsymbol{h}_{z_A}^{(l)}]$. The multi-head self-attention (MHA) in the $(l + 1)$-th layer will be

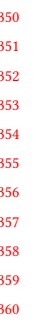
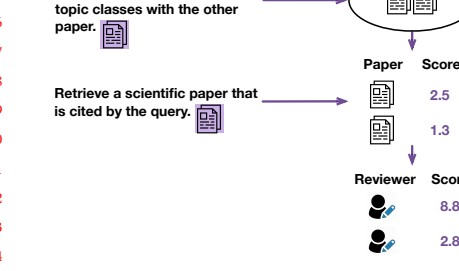

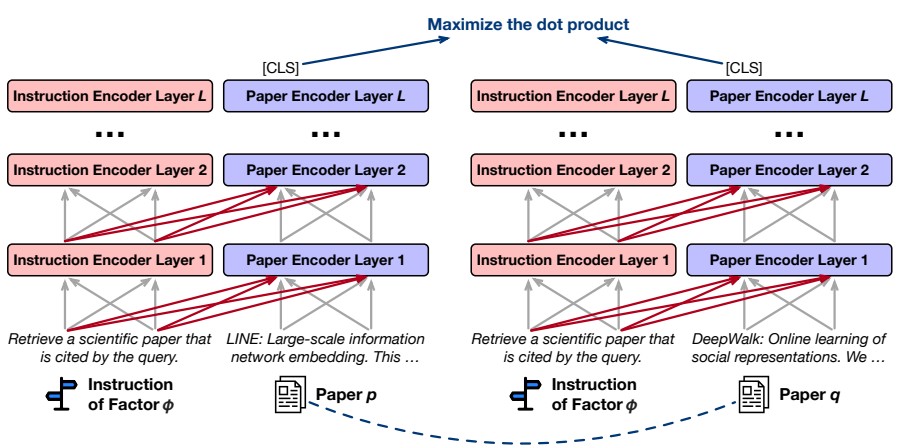

**Figure 2: The CHAIN-OF-FACTORS matching process.**

**Figure 3: Pre-training the instruction encoder and the paper encoder to learn factor-aware paper representations.**

calculated as follows:

$$\text{MHA}(H_{i_\phi}^{(l)}) = \|_{u=1}^{U} \text{head}_u(H_{i_\phi}^{(l)}),$$

$$\text{where} \quad \text{head}_u(H_{i_\phi}^{(l)}) = \text{softmax}\left(\frac{Q_u^{(l)} K_u^{(l)\top}}{\sqrt{d/U}}\right) \cdot V_u^{(l)}, \quad (5)$$

$$Q_u^{(l)} = H_{i_\phi}^{(l)} W_{Q,u}^{(l)}, \quad K_u^{(l)} = H_{i_\phi}^{(l)} W_{K,u}^{(l)}, \quad V_u^{(l)} = H_{i_\phi}^{(l)} W_{V,u}^{(l)}.$$

With the MHA mechanism, the encoding process of the $(l+1)$-th layer will be:

$$\widehat{H}_{i_\phi}^{(l)} = \text{LN}(H_{i_\phi}^{(l)} + \text{MHA}(H_{i_\phi}^{(l)})),$$

$$H_{i_\phi}^{(l+1)} = \text{LN}(\widehat{H}_{i_\phi}^{(l)} + \text{FFN}(\widehat{H}_{i_\phi}^{(l)})), \quad (6)$$

where $\text{LN}(\cdot)$ is the layer normalization operator [3] and $\text{FFN}(\cdot)$ is the position-wise feed-forward network [52].

**Paper Encoding.** After instruction encoding, the paper encoder $\text{Enc}_p(\cdot)$ takes instruction representations as contexts to guide the encoding process of each paper $p = w_1 w_2 ... w_B$. Specifically, $\text{Enc}_p(\cdot)$ has the same number of (i.e., 12) layers as $\text{Enc}_i(\cdot)$, and the encoding process of $\text{Enc}_p(\cdot)$'s $(l+1)$-th layer incorporates the instruction inputs from $\text{Enc}_i(\cdot)$'s corresponding layer (i.e., $H_{i_\phi}^{(l)}$) into its MHA calculation. Formally, we define:

$$H_p^{(l)} = [h_{w_1}^{(l)}, h_{w_2}^{(l)}, ..., h_{w_B}^{(l)}],$$

$$\widetilde{H}_p^{(l)} = H_{i_\phi}^{(l)} \| H_p^{(l)} = [h_{z_1}^{(l)}, h_{z_2}^{(l)}, ..., h_{z_A}^{(l)}, h_{w_1}^{(l)}, h_{w_2}^{(l)}, ..., h_{w_B}^{(l)}]. \quad (7)$$

Taking instructional contexts into account, we calculate the following asymmetric MHA [57]:

$$\text{MHA}_{asy}(H_p^{(l)}, \widetilde{H}_p^{(l)}) = \|_{u=1}^{U} \text{head}_u(H_p^{(l)}, \widetilde{H}_p^{(l)}),$$

$$\text{where} \quad \text{head}_u(H_p^{(l)}, \widetilde{H}_p^{(l)}) = \text{softmax}\left(\frac{Q_u^{(l)} \widetilde{K}_u^{(l)\top}}{\sqrt{d/U}}\right) \cdot \widetilde{V}_u^{(l)}, \quad (8)$$

$$Q_u^{(l)} = H_p^{(l)} W_{Q,u}^{(l)}, \quad \widetilde{K}_u^{(l)} = \widetilde{H}_p^{(l)} W_{K,u}^{(l)}, \quad \widetilde{V}_u^{(l)} = \widetilde{H}_p^{(l)} W_{V,u}^{(l)}.$$

The key differences between Eq. (8) and Eq. (5) are highlighted in blue. With the asymmetric MHA mechanism, the paper encoding

process of the $(l+1)$-th layer will be:

$$\widehat{H}_p^{(l)} = \text{LN}(H_p^{(l)} + \text{MHA}_{asy}(H_p^{(l)}, \widetilde{H}_p^{(l)})),$$

$$H_p^{(l+1)} = \text{LN}(\widehat{H}_p^{(l)} + \text{FFN}(\widehat{H}_p^{(l)})). \quad (9)$$

The final instruction-guided representation of $p$ is the output embedding of its [CLS] token after the last layer. In other words, $g(p|\phi) = h_{[\text{CLS}]}^{(12)}$.

**Summary.** To give an intuitive summary of the encoding process, as shown in Figure 3, the instruction $i_\phi$ serves as the context of the paper $p$ (via attention illustrated by the red arrows), making the final paper representation aware of the corresponding factor $\phi$. Conversely, the paper does *not* serve as the context of the instruction because we want the semantic meaning of the instruction to be stable and not affected by a specific paper. The parameters of the two encoders $\text{Enc}_i(\cdot)$ and $\text{Enc}_p(\cdot)$ are shared during training. All three factors also share the same $\text{Enc}_i(\cdot)$ and the same $\text{Enc}_p(\cdot)$ so that the model can carry common knowledge learned from pre-training data of different factors.

## 3.3 Model Training

In this section, we introduce the data and objective used to pre-trained the instruction-guided paper encoder $g(\cdot|\phi)$.

**Pre-training Data.** For the semantic factor, each submission $p$ is treated as a "query" and each paper $q$ in a reviewer's publication profile is viewed as a "document". In this case, $g(\cdot|\phi)$ should be learned to maximize the inner product of an ad-hoc query and its semantically relevant document in the vector space. To facilitate this, we adopt the Search dataset from the SciRepEval benchmark [47] to pre-train our model, where the queries are collected from an academic search engine, and the relevant documents are derived from large-scale user click-through data.

For the topic factor, $g(\cdot|\phi)$ should be trained to maximize the inner product of two papers $p$ and $q$ if they share common research fields. We utilize the MAPLE benchmark [63] as pre-training data in which millions of scientific papers are tagged with their fine-grained fields-of-study from the Microsoft Academic Graph [48].

For example, for CS papers in MAPLE, there are over 15K fine-grained research fields, and each paper is tagged with about 6 fields on average. Such data are used to derive topically relevant paper pairs to pre-train $g(\cdot|\phi)$.

For the citation factor, $g(\cdot|\phi)$ should be learned to maximize the inner product of two papers $p$ and $q$ if $p$ cites $q$. Following [9, 36], we leverage a large collection of citation triplets $(p, q^+, q^-)$ constructed by Cohan et al. [9], where $p$ cites $q^+$ but does not cite $q^-$, to pre-train $g(\cdot|\phi)$.

One can refer to Appendix A.1.1 for more details of the pre-training data.

**Pre-training Objective.** For all three factors, each sample from their pre-training data can be denoted as $(p, q^+, q_1^-, q_2^-, ..., q_T^-)$, where $q^+$ is relevant to $p$ (i.e., when $\phi$ = semantic, $q^+$ is clicked by users in a search engine given the search query $p$; when $\phi$ = topic, $q^+$ shares fine-grained research fields with $p$; when $\phi$ = citation, $q^+$ is cited by $p$) and $q_t^-$ ($t = 1, 2, ..., T$) are irrelevant to $p$. Given a factor $\phi$ and its training sample $(p, q^+, q_1^-, q_2^-, ..., q_T^-)$, we randomly pick an instruction $i_\phi$ corresponding to $\phi$. Then, using the instruction encoder $\text{Enc}_i(\cdot)$ and the paper encoder $\text{Enc}_p(\cdot)$, we can obtain $g(p|\phi)$, $g(q^+|\phi)$, $g(q_1^-|\phi)$, ..., and $g(q_T^-|\phi)$, and we adopt a contrastive loss [35] to train our model:

$$\mathcal{J} = -\log \frac{\exp(g(p|\phi)^\top g(q^+|\phi))}{\exp(g(p|\phi)^\top g(q^+|\phi)) + \sum_{t=1}^{T} \exp(g(p|\phi)^\top g(q_t^-|\phi))}. \quad (10)$$

The overall pre-training objective is:

$$\min_{\text{Enc}_i(\cdot), \text{Enc}_p(\cdot)} \sum_\phi \sum_{(p, q^+, q_1^-, q_2^-, ..., q_T^-)} \mathcal{J}. \quad (11)$$

Note that our training paradigm is different from prefix/prompt-tuning [25, 27, 28]. To be specific, prefix/prompt-tuning freezes the backbone language model and optimizes the prefix/prompt part only, and its major goal is a more *efficient* language model tuning paradigm. By contrast, we train the instruction encoder and the paper encoder simultaneously, aiming for a more *effective* unified model to obtain factor-aware text representations.

**Hard Negatives.** Cohan et al. [9] show that a combination of easy negatives and hard negatives boosts the performance of their contrastive learning model. Following their idea, we take in-batch negatives [22] as easy negatives and adopt the following strategies to find hard negatives: when $\phi$ = semantic, $q_t^-$ is a hard negative if it is shown to users but not clicked given the query $p$; when $\phi$ = topic, $q_t^-$ is a hard negative if it shares the same venue but does not share any fine-grained field with $p$; when $\phi$ = citation, $q_t^-$ is a hard negative if $q^+$ cites $q_t^-$ but $p$ does not cite $q_t^-$.

## 4 EXPERIMENTS

### 4.1 Setup

*4.1.1 Evaluation Datasets.* Collecting the ground truths of paper-reviewer relevance is challenging. Some related studies [1, 41, 43] can fortunately access actual reviewer bidding data in previous conferences where reviewers self-report their expertise on certain papers, but such confidential information cannot be released, so the used datasets are not publicly available. Alternatively, released benchmark datasets [21, 33, 65] gather paper-reviewer relevance judgments from annotators with domain expertise. In our experiments, we adopt the latter solution and consider four publicly

**Table 1: Dataset Statistics.**

| Dataset | #Papers | #Reviewers | #Annotated $(p, r)$ Pairs | Conference(s) |
|---|---|---|---|---|
| NIPS [33] | 34 | 190 | 393 | NIPS 2006 |
| SciRepEval [47] | 107 | 661 | 1,729 | NIPS 2006, ICIP 2016 |
| SIGIR [21] | 73 | 189 | 13,797 | SIGIR 2007 |
| KDD | 174 | 737 | 3,480 | KDD 2020 |

available datasets covering diverse domains, including machine learning, computer vision, information retrieval, and data mining.

- **NIPS [33]** is a pioneering benchmark dataset for evaluating paper-reviewer matching. It consists of expertise relevance judgements between 34 papers accepted by NIPS 2006 and 190 reviewers. Annotations were done by 9 researchers from the NIPS community, and the score of each annotated paper-reviewer pair can be "3" (very relevant), "2" (relevant), "1" (slightly relevant), or "0" (irrelevant). Note that for each paper, the annotators only judge its relevance with a subset of reviewers, and a total of 393 paper-reviewer pairs are annotated.

- **SciRepEval [47]** is a comprehensive benchmark for scientific document representation learning. Its paper-reviewer matching dataset combines the annotation effort from multiple sources [29, 33, 65]. Specifically, Liu et al. [29] added relevance scores of 766 paper-reviewer pairs to the NIPS dataset to mitigate its annotation sparsity; Zhao et al. [65] provided 694 paper-reviewer relevance ratings for the ICIP 2016 conference. The combined dataset still adopts the "0"-"3" rating scale.

- **SIGIR [21]** contains 73 papers accepted by SIGIR 2007 and 189 prospective reviewers. Instead of annotating each specific paper-reviewer pair, the dataset constructors assign one or more aspects of information retrieval (e.g., "Evaluation", "Web IR", and "Language Models", with 25 candidate aspects in total) to each paper and each reviewer. Then, the relevance between a paper and a reviewer is determined by their aspect-level similarity. In our experiments, to align with the rating scale in NIPS and SciRepEval, we discretize the Jaccard similarity between a paper's aspects and a reviewer's aspects to map their relevance to "0", "1", "2", or "3".

- **KDD** is a new dataset introduced in this paper annotated by us. Our motivation for constructing it is to contribute a paper-reviewer matching dataset with _more recent data mining_ papers. The dataset contains relevance scores of 3,480 paper-reviewer pairs between 174 papers accepted by KDD 2020 and 737 prospective reviewers. Annotations were done by 5 data mining researchers, following the "0"-"3" rating scale. More details on the dataset construction process can be found in Appendix A.1.2.

Following [33, 47], we consider two different task settings: In the *Soft* setting, reviewers with a score of "2" or "3" are considered as relevant; in the *Hard* setting, only reviewers with a score of "3" are viewed as relevant. Dataset statistics are summarized in Table 1.

*4.1.2 Compared Methods.* We compare CoF with both classical paper-reviewer matching baselines and pre-trained language models considering different factors.

- **Author-Persona-Topic Model (APT200) [33]** is a **topic** model specifically designed for paper-reviewer matching. It augments the generative process of LDA with authors and personas, where

**Table 2: P@5 scores on the NIPS dataset. Bold: the highest score. \*: CoF is significantly better than this method with p-value < 0.05. \*\*: CoF is significantly better than this method with p-value < 0.01.** Red , Yellow , Blue : models mainly focusing on the semantic , topic , and citation factors, respectively. Scores of APT200, RWR, and Common Topic Model are reported in [33], [29], and [1], respectively.

| | NIPS [33] | | | |
|---|---|---|---|---|
| | Soft P@5 | Hard P@5 | P@5 defined in [29] | P@5 defined in [1] |
| APT200 [33] | 41.18** | 20.59** | – | – |
| TPMS [7] | 49.41** | 22.94** | 50.59** | 55.15** |
| RWR [29] | – | 24.1** | 45.3** | – |
| Common Topic Model [1] | – | – | – | 56.6** |
| SciBERT [6] | 47.06** | 21.18** | 49.61** | 52.79** |
| SPECTER [9] | 52.94** | 25.29** | 53.33** | 58.68** |
| SciNCL [36] | 54.12** | 27.06** | 54.71** | 59.85** |
| COCO-DR [59] | 54.12** | 25.29** | 54.51** | 59.85** |
| SPECTER 2.0 CLF [47] | 52.35** | 24.71** | 53.33** | 58.09** |
| SPECTER 2.0 PRX [47] | 53.53** | 27.65 | 54.71** | 59.26** |
| CoF | **55.68** | **28.24** | **56.41** | **61.42** |

each author can write papers under one or more personas represented as distributions over hidden topics.

- **Toronto Paper Matching System (TPMS) [7]** focuses on the **semantic** factor and defines paper-reviewer relevance as the tf–idf similarity between them.
- **Random Walk with Restart (RWR) [29]** mainly considers the **topic** factor for paper-reviewer matching. It constructs a graph with reviewer-reviewer edges (representing co-authorship) and submission-reviewer edges (derived from topic-based similarity after running LDA). Then, the model conducts random walk with restart on the graph to calculate submission-reviewer proximity.
- **Common Topic Model [1]** is an embedding-based **topic** model specifically designed for paper-reviewer matching. It jointly models the common topics of submissions and reviewers by taking the word2vec embeddings [32] as input.
- **SciBERT [6]** is a PLM trained on scientific papers following the idea of BERT (i.e., taking masked language modeling and next sentence prediction as pre-training tasks).
- **SPECTER [9]** is a scientific PLM initialized from SciBERT and trained on **citation** links between papers.
- **SciNCL [36]** is also a scientific PLM initialized from SciBERT and trained on **citation** links. It improves the hard negative sampling strategy of SPECTER.
- **COCO-DR [59]** is a PLM trained on MS MARCO [4] for zero-shot dense information retrieval. We view COCO-DR as a representative PLM baseline focusing on the **semantic** factor.
- **SPECTER 2.0 [47]** is a PLM trained on a wide range of scientific literature understanding tasks. It adopts the architecture of adapters [39] for multi-task learning, so there are different model variants. We consider two variants in our experiments: **SPECTER 2.0 PRX** is mainly trained on **citation** prediction and same author prediction tasks. It is evaluated for paper-reviewer matching in [47]. **SPECTER 2.0 CLF** is mainly trained on classification tasks. Although it is not evaluated for paper-reviewer matching in [47], we view it as a representative PLM baseline focusing on the **topic** factor.

Implementation details and hyperparameter configurations of the baselines and CoF can be found in Appendices A.2.1 and A.2.2.

*4.1.3 Evaluation Metrics.* Following [33, 47], we adopt P@5 and P@10 as evaluation metrics. For each submission paper $p$, let $\mathcal{R}_p$ denote the set of candidate reviewers that have an annotated relevance score with $p$; let $r_{p,k}$ denote the reviewer ranked $k$-th in $\mathcal{R}_p$ according to $f(p, r)$. Then, the P@K scores ($K = 5$ and $10$) under the Soft and Hard settings are defined as:

$$\text{Soft P@}K = \frac{1}{|\mathcal{P}|} \sum_{p \in \mathcal{P}} \frac{\sum_{k=1}^{K} \mathbf{1}(\text{score}(p, r_{p,k}) \geq 2)}{K},$$

$$\text{Hard P@}K = \frac{1}{|\mathcal{P}|} \sum_{p \in \mathcal{P}} \frac{\sum_{k=1}^{K} \mathbf{1}(\text{score}(p, r_{p,k}) = 3)}{K}. \quad (12)$$

Here, $\mathbf{1}(\cdot)$ is the indicator function; $\text{score}(p, r)$ is the annotated relevance score between $p$ and $r$.

## 4.2 Performance Comparison

Tables 2 and 3 show the performance of compared methods on the four datasets. We are unable to find a publicly available implementation of APT200, RWR, and Common Topic Model, so we put their reported performance on the NIPS dataset [1, 29, 33] into Table 2. Note that in [29] and [1], the definitions of (Soft) P@K are slightly different from that in Eq. (12). To be specific,

$$\text{P@}K \text{ defined in [29]} = \frac{1}{|\mathcal{P}|} \sum_{p \in \mathcal{P}} \frac{\sum_{k=1}^{K} \text{score}(p, r_{p,k})}{3K},$$

$$\text{P@}K \text{ defined in [1]} = \frac{1}{|\mathcal{P}|} \sum_{p \in \mathcal{P}} \frac{\sum_{k=1}^{K} \mathbf{1}(\text{score}(p, r_{p,k}) \geq 2)}{\min\{K, |\mathcal{R}_p|\}}. \quad (13)$$

To compare with the numbers reported in [29] and [1] on NIPS, we also calculate the P@K scores following these two alternative definitions and show them in Table 2.

In Tables 2 and 3, to show statistical significance, we run CoF 3 times and conduct a two-tailed Z-test to compare CoF with each baseline. The significance level is also marked in the two tables. We can observe that: (1) On the NIPS dataset, CoF consistently achieves the best performance in terms of all shown metrics. In all but one of the cases, the improvement is significant with p-value < 0.01. On SciRepEval, SIGIR, and KDD, we calculate the average of the four metrics (i.e., {Soft, Hard} × {P@5, P@10}). In terms of the average metric, CoF consistently and significantly outperforms all baselines. If we check each metric separately, CoF achieves the highest score in 9 out of 12 columns. (2) PLM baselines always outperform classical paper-reviewer matching baselines considering the same factor. This rationalizes our motivation to unify all factors with a PLM-based framework.

## 4.3 Ablation Study

The key technical novelty of CoF is twofold: (1) we use instruction tuning to learn factor-specific representations during pre-training, and (2) we exploit chain-of-factors matching during inference. Now we demonstrate the contribution of our proposed techniques through a comprehensive ablation study. To be specific, we examine the following ablation versions:

**Table 3: P@5 and P@10 scores on the SciRepEval, SIGIR, and KDD datasets. Bold, ∗, ∗∗, Red, Yellow, and Blue : the same meaning as in Table 2.**

| | SciRepEval [47] | | | | | SIGIR [21] | | | | | KDD | | | | |
|---|---|---|---|---|---|---|---|---|---|---|---|---|---|---|---|
| | Soft P@5 | Soft P@10 | Hard P@5 | Hard P@10 | Average | Soft P@5 | Soft P@10 | Hard P@5 | Hard P@10 | Average | Soft P@5 | Soft P@10 | Hard P@5 | Hard P@10 | Average |
| TPMS [7] | 62.06** | 53.74** | 31.40** | 24.86** | 43.02** | 39.73** | 38.36** | 17.81** | 17.12** | 28.26** | 17.01** | 16.78** | 6.78** | 7.24** | 11.95** |
| SciBERT [6] | 59.63** | 54.39** | 28.04** | 24.49** | 41.64** | 34.79** | 34.79** | 14.79** | 15.34** | 24.93** | 28.51** | 27.36** | 12.64** | 12.70** | 20.30** |
| SPECTER [9] | 65.23** | **56.07** | 32.34** | 25.42 | 44.77** | 39.73** | 40.00** | 16.44** | 16.71** | 28.22** | 34.94** | 30.52** | 15.17** | **13.28** | 23.48** |
| SciNCL [36] | 66.92** | 55.42** | 34.02* | 25.33 | 45.42** | 40.55** | 39.45** | 17.81** | 17.40* | 28.80** | 36.21** | 30.86** | 15.06** | 12.70** | 23.71** |
| COCO-DR [59] | 65.05** | 55.14** | 31.78** | 24.67** | 44.16** | 40.00** | 40.55* | 16.71** | 17.53 | 28.70** | 35.06** | 29.89** | 13.68** | 12.13** | 22.69** |
| SPECTER 2.0 CLF [47] | 64.49** | 55.23** | 31.59** | 24.49** | 43.95** | 39.45** | 38.63** | 16.16** | 16.30** | 27.64** | 34.37** | 30.63** | 14.48** | 12.64** | 23.03** |
| SPECTER 2.0 PRX [47] | 66.36** | 55.61** | 34.21 | **25.61** | 45.45** | 40.00** | 38.90** | 19.18** | 16.85** | 28.73** | 37.13 | 31.03 | 15.86** | 13.05* | 24.27* |
| CoF | **68.47** | 55.89 | **34.52** | 25.33 | **46.05** | **45.57** | **41.69** | **22.47** | **17.76** | **31.87** | **37.63** | **31.09** | **16.13** | 13.08 | **24.48** |

**Table 4: Average metrics of CoF and its ablation versions on NIPS, SIGIR, and KDD.**

| | NIPS | SIGIR | KDD |
|---|---|---|---|
| CoF  $(\mathbb{S} \to \mathbb{T} \to \mathbb{S} + \mathbb{T} + \mathbb{C})$ | 50.44 | **31.87** | **24.48** |
| No-Instruction | 49.52** | 27.67** | 24.07** |
| $\mathbb{S}$ | 50.29 | 28.07** | 24.05** |
| $\mathbb{T}$ | 49.98 | 28.69** | 24.11* |
| $\mathbb{C}$ | 50.31 | 28.81** | 24.20* |
| $\mathbb{S} + \mathbb{T} + \mathbb{C}$ | **50.55** | 28.63** | 24.26* |
| $\mathbb{S} \to \mathbb{T} \to \mathbb{C}$ | 50.11 | 31.79 | 24.36 |

- **No-Instruction** takes all pre-training data to train one paper encoder without using instructions. In this way, the model can only output one factor-agnostic embedding for each paper.
- The model can be pre-trained on data from all three factors but only consider one factor during inference. This yields 3 ablation versions, denoted as $\mathbb{S}$, $\mathbb{T}$, and $\mathbb{C}$, considering semantic, topic, and citation information, respectively.
- The model can consider all three factors during inference without chain-of-factors matching. In this case, it directly uses

$$f(p, r) = \sum_{q \in Q_r} \Big( f_{\text{semantic}}(p, q) + f_{\text{topic}}(p, q) + f_{\text{citation}}(p, q) \Big)$$

as the criteria to rank all candidate reviewers, and we denote this ablation version as $\mathbb{S} + \mathbb{T} + \mathbb{C}$.
- The model can adopt a chain-of-factors matching strategy but only utilize citation information in the last step of the chain. We denote this variant as $\mathbb{S} \to \mathbb{T} \to \mathbb{C}$.

Table 4 compares the full CoF model (i.e., $\mathbb{S} \to \mathbb{T} \to \mathbb{S} + \mathbb{T} + \mathbb{C}$) with aforementioned ablation versions on NIPS, SIGIR, and KDD. We can see that: (1) the full model always significantly outperforms No-Instruction, indicating the importance of our proposed instruction-aware pre-training step. (2) On SIGIR and KDD, the full model is significantly better than $\mathbb{S}$, $\mathbb{T}$, $\mathbb{C}$ and $\mathbb{S} + \mathbb{T} + \mathbb{C}$. This highlights the benefits of considering multiple factors *and* adopting a chain-of-factors matching strategy during inference, corresponding to the two technical contributions of CoF. (3) The full model is consistently better than $\mathbb{S} \to \mathbb{T} \to \mathbb{C}$, but the gap is not significant. In particular, on SIGIR, there is a very clear margin between models with chain-of-factors matching and those without.

## 4.4 Effectiveness in Each Factor

One may suspect that some baselines are more powerful than CoF in a certain factor, but finally underperform CoF in paper-reviewer

**Table 5: Performance of compared models in three tasks related to semantic, topic, and citation factors, respectively, on KDD. All three tasks require a model to rank 100 candidates (1 relevant and 99 irrelevant) for each query. We report the mean rank of the relevant candidate achieved by each model (the lower the better).**

| | Semantic ($\mathbb{S}$) | Topic ($\mathbb{T}$) | Citation ($\mathbb{C}$) |
|---|---|---|---|
| SciBERT [6] | 10.88** | 25.52** | 19.47** |
| SPECTER [9] | 3.37** | 7.90** | 6.12** |
| SciNCL [36] | 1.40** | 6.05** | 5.35** |
| COCO-DR [59] | 2.55** | 7.34** | 9.80** |
| SPECTER 2.0 CLF [47] | 4.41** | 12.56** | 9.69** |
| SPECTER 2.0 PRX [47] | 1.33* | 6.11** | 4.75** |
| CoF | **1.21** | **3.02** | 3.97 |

matching just because CoF takes more factors into account and adopts chain-of-factors matching. To dispel such misgivings, we examine the performance of compared methods in three tasks – semantic retrieval, topic classification, and citation prediction – corresponding to the three factors, respectively. Specifically, for each submission paper $p$ in the KDD dataset[2], we sample 100 candidates among which only one is "relevant" to $p$ and the other 99 are "irrelevant". Here, the meaning of "relevant" depends on the examined task. For topic classification, we sample one of $p$'s fields, and the "relevant" candidate is the name of that field; the "irrelevant" candidates are names of other randomly sampled fields. For citation prediction, we select one of the papers cited by $p$ as the "relevant" candidate; the "irrelevant" candidates are chosen from candidate reviewers' previous papers not cited by $p$. For semantic retrieval, we follow [19] to conduct a title-to-abstract retrieval task, where the query is $p$'s title, the "relevant" candidate is $p$'s abstract, and the "irrelevant" candidates are sampled from other papers' abstracts. Note that for the topic classification task, the instructions used by CoF for paper-reviewer matching are no longer suitable, so we adopt a new instruction "Tag a scientific paper with relevant scientific topic classes.".

For each task, we ask compared models to rank the 100 candidates for each submission $p$ and calculate the mean rank of the "relevant" candidate. A perfect model should achieve a mean rank of 1, and a random guesser will get an expected mean rank of 50.5.

---

[2]We use the KDD dataset for experiments in Sections 4.4 and 4.5 because the required information of each paper (e.g., venue, year, references, fields) is stored when we construct the dataset. By contrast, such information is largely missing in the other three datasets.

Table 5 shows the performance of compared models in the three tasks, where CoF consistently performs the best. This observation proves that the reasons why CoF can outperform baselines in paper-reviewer matching are twofold: (1) CoF jointly considers three factors in a chain manner (the benefit of which has been shown in Table 4), and (2) CoF indeed improves upon the baselines in *each* of the three factors.

## 4.5 Effect of Reviewers' Publication Profile

How to form each reviewer's publication profile may affect model performance. Shall we include all papers written by a reviewer or set up some criteria? Here, we explore the effect of three intuitive criteria. (1) *Time span*: What if we include papers published in the most recent $Y$ years only (because earlier papers may have diverged from reviewers' current interests)? For example, for the KDD 2020 conference, if $Y = 5$, then we only put papers published during 2015-2019 into reviewers' publication profile. Figure 4(a) shows the performance of CoF with $Y = 1, 2, 5, 10$, and 20. We observe that including more papers is always beneficial, but the performance starts to converge at $Y = 10$. (2) *Venue*: What if we include papers published in top venues only? Figure 4(b) compares the performance of using all papers written by the reviewers with that of using papers published in "top conferences" only. Here, "top conferences" refer to the 75 conferences listed on CSRankings[3] in 2020 (with KDD included). The comparison implies that papers not published in top conferences still have a positive contribution to characterize reviewers' expertise. (3) *Rank in the author list*: What if we include each reviewer's first-author and/or last-author papers only (because these two authors often contribute most to the paper according to [11])? Figure 4(b) also shows the performance of using each reviewer's first-author papers, last-author ones, and the union of them. Although the union is evidently better than either alone, it is still obviously behind using all papers. To summarize our findings, when the indication from reviewers is not available, putting the *whole* set of their papers into their publication profile is almost always helpful. This is possibly because our chain-of-factors matching strategy enables coarse-to-fine filtering of irrelevant papers, making the model more robust towards noises.

## 5 RELATED WORK

**Paper-Reviewer Matching.** Following the logic of the entire paper, we divide previous paper-reviewer matching methods according to the factor they consider. In earlier times, *semantic*-based approaches use bag-of-words representations, such as tf–idf vectors [7, 16, 58] and keywords [40, 51] to describe each submission paper and each reviewer. As a key technique in classical information retrieval, probabilistic language models have also been utilized in expert finding [5]. More recent semantic-based methods have started to employ context-free word embeddings [60, 64] for representation. *Topic*-based approaches leverage topic models such as Latent Semantic Indexing [14, 26], Probabilistic Latent Semantics Analysis [10, 21], and Latent Dirichlet Allocation (and its variants) [18, 24, 29] to infer each paper/reviewer's topic distribution. This idea is recently improved by exploiting embedding-enhanced topic models [1, 41]. Inspired by the superiority of contextualized language models to context-free representations, recent studies [47, 50] apply scientific

---

3https://csrankings.org/

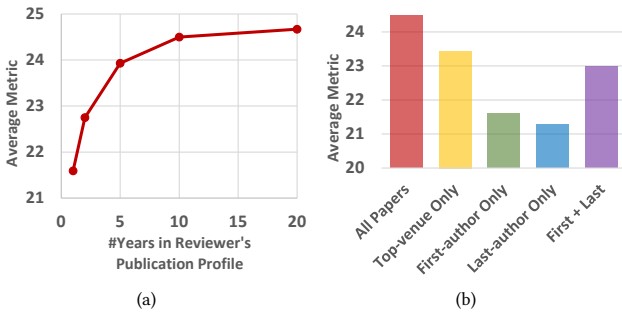

(a)        (b)

**Figure 4: (a) Performance of CoF with different time spans within which the reviewers' previous papers are considered. (b) Performance of CoF with different criteria to construct reviewers' publication profile.**

PLMs such as SPECTER [9], SciNCL [36], and SPECTER 2.0 [47] to perform paper-reviewer matching. These PLMs are pre-trained on a large amount of *citation* information between papers. For a more complete discussion of paper-reviewer matching studies, one can refer to a recent survey [65]. Note that most of the aforementioned approaches take only one factor into account, resulting in an incomprehensive estimation of the paper-reviewer relevance. In comparison, CoF jointly considers the semantic, topic, and citation factors with a unified model.

**Instruction Tuning.** Training (large) language models to follow instructions on many tasks has been extensively studied [8, 37, 45, 53, 54]. However, these instruction-tuned language models mainly adopt a decoder-only or encoder-decoder architecture with billions of parameters, aiming at generation tasks and hard to adapt for paper-reviewer matching. Moreover, the major goal of these studies is to facilitate zero-shot or few-shot transfer to new tasks rather than learning task-aware representations. Recently, Asai et al. [2] propose to utilize task-specific instructions for information retrieval; Zhang et al. [62] further explore instruction tuning in various scientific literature understanding tasks such as paper classification and link prediction. However, unlike CoF, these models do not fuse signals from multiple tasks/factors during inference, and paper-reviewer matching is not their target task.

## 6 CONCLUSIONS

In this work, we present a CHAIN-OF-FACTORS framework that jointly considers semantic, topic, and citation signals in a step-by-step, coarse-to-fine manner for paper-reviewer matching. We propose an instruction-guided paper encoding process to learn factor-aware text representations so as to model paper-reviewer relevance of different factors. Such a process is facilitated by pre-training an instruction encoder and a paper encoder with a contextualized language model backbone. Experimental results validate the efficacy of our CoF framework on four datasets across various fields. Further ablation studies explain the major reasons why CoF is superior to the baselines: (1) CoF comprehensively considers three factors rather than just one, (2) CoF chains three factors to facilitate step-by-step, coarse-to-fine relevant paper selection instead of merging all factors in one step, and (3) CoF improves upon the baselines in each of the factors. We also conduct analyses on how the composition of each reviewer's publication profile will affect the paper-reviewer matching performance.

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

# A APPENDIX

## A.1 Datasets

*A.1.1 Pre-training Data.* We have briefly introduced the pre-training data in Section 3.3. Here are more details.

- **Search** from **SciRepEval [47]**[4] is used for the semantic factor. It has over 528K queries. For each query, a list of documents is given, and the score between the query and each document falls into $\{0, 1, ..., 14\}$, reflecting how often the document is clicked by users given the query. We treat a document as relevant if it has a non-zero score with the query. Other documents in the list are viewed as hard negatives.
- **CS-Journal** from **MAPLE [63]**[5] is used for the topic factor. It has more than 410K papers published in top CS journals from 1981 to 2020. We choose CS-Journal instead of CS-Conference from MAPLE for pre-training so as to mitigate data leakage because the four evaluation datasets are all constructed from previous conference papers. In CS-Journal, each paper is tagged with its relevant fields. There are over 15K fine-grained fields organized into a 5-layer hierarchy [46]. Two papers are treated as relevant if they share at least one field at Layer 3 or deeper.
- **Citation Prediction Triplets [9]**[6] are used for the citation factor. There are more than 819K paper triplets $(p, q^+, q^-)$, where $p$ cites $q^+$, $q^+$ cites $q^-$, but $p$ does not cite $q^-$.

*A.1.2 Construction of the KDD Dataset.* We rely on the Microsoft Academic Graph (MAG) [48] to extract each paper's title, abstract, venue, and author(s). The latest KDD conference available in our downloaded MAG is KDD 2020. Therefore, we first retrieve all KDD 2020 papers from MAG as potential "submission" papers. Then, we select those researchers meeting the following two criteria as candidate reviewers: (1) having published at least 1 KDD paper during 2018-2020, and (2) having published at least 3 papers in "top conferences". Consistent with the definition in Section 4.5, "top-conferences" refer to the 75 conferences listed on CSRankings in 2020, including KDD. Guided by our observations in Section 4.5, for each candidate reviewer $r$, we include *all* of its papers published in 2019 or earlier to form its publication profile $Q_r$. Next, we randomly sample about 200 papers from KDD 2020. For each sampled paper, we select 20 candidate reviewers for annotation. We do our best to ensure that conflict-of-interest reviewers (e.g., authors and their previous collaborators) are not selected. To reduce the possibility that none of the selected reviewers is relevant to the paper (which makes the paper useless in evaluation), reviewers sharing a higher TPMS score [7] with the paper are more likely

to be selected for annotation. Finally, we invite 5 annotators to independently rate each pair of (paper, selected reviewer) according to the "0"-"3" relevance scheme. The final score between a paper and a reviewer is the average rating from the annotators rounded to the nearest integer. We remove papers that: (1) do not have any selected reviewer with an annotated relevance score greater than or equal to "2" or (2) annotators are not able to judge its relevance to some candidate reviewers, resulting in 174 papers in the final dataset. On average, each paper in our KDD dataset has 2.10 reviewers with a relevance score of "3", 3.05 reviewers with a score of "2", 6.32 reviewers with a score of "1", and 8.53 reviewers with a score of "0".

## A.2 Implementation Details

*A.2.1 Baselines.* We use the following implementation/checkpoint of each baseline:

- **TPMS**: https://github.com/niharshah/goldstandard-reviewer-paper-match/blob/main/scripts/tpms.py
- **SciBERT**: https://huggingface.co/allenai/scibert_scivocab_uncased
- **SPECTER**: https://huggingface.co/allenai/specter
- **SciNCL**: https://huggingface.co/malteos/scincl
- **COCO-DR**: https://huggingface.co/OpenMatch/cocodr-base-msmarco
- **SPECTER 2.0**: https://huggingface.co/allenai/specter2

For PLM baselines, we follow [47] and adopt the average of top-3 values to aggregate paper-paper relevance to paper-reviewer relevance.

*A.2.2 CoF.* The maximum input sequence lengths of instructions and papers are set as 32 tokens and 256 tokens, respectively. We train the model for 20 epochs with a peak learning rate of 3e-4 and a weight decay of 0.01. The AdamW optimizer [31] is used with $(\beta_1, \beta_2) = (0.9, 0.999)$. The batch size is 32. For each training sample, we create one hard negative and combine it with easy in-batch negatives for contrastive learning.

---

[4]https://huggingface.co/datasets/allenai/scirepeval/viewer/search
[5]https://github.com/yuzhimanhua/MAPLE
[6]https://huggingface.co/datasets/allenai/scirepeval/viewer/cite_prediction

