# OpenReview forum: "Chain-of-Factors Paper-Reviewer Matching"
_ACM.org/TheWebConf/2025/Conference — WWW 2025 Oral_

### Official Review · Reviewer_sGqq · 2024-11-25

**Novelty:** 5
**Technical Quality:** 6

**Review:**

The paper addresses the problem of paper-reviewer matching by proposing a combination of semantic, topic, and citation factors via instruction-tuning of LMs.

Strengths:

S1: The authors make an explicit effort to show the relevance to the WebConf and, while not straightforward from the title-abstract, I am not concerned about the paper's scope.

S2: While a bit over-formelized in the model, the paper is quite easy to follow and its readability is good.

S3: The experiments are quite through with many baseline methods and a decent amount of datasets and analysis.

S4: New dataset


Weaknesses:

W1: Problem definition and model: I do not like the problem definition as the core goal of this work is assessing the relevance of a given paper to a given reviewer. The authors made a clear argument why the ranking and constraints are outside the scope of this work; yet, the problem definition does not reflect that. There's some discussion about it (and results) but it is not clear what is the role of the number of papers a potential reviewer has. There are some factors that are taken into account in more than one dimention, e.g., if the semantics are similar the reviewers is also more likely to work on the same topic, how is this addressed? Are the topics well-defined? It seems like the citation factor is also topic-based, is that really the case? I am missing some additional clarifications about the instruction tuning? Is it like prompt? I am also not sure that the very technical details of the encoding are relevant to understand the paper and hurt the readbility which is quite good.

W2: Datasets and experiments: The benchmark datasets (top 3) are quite old, is this still representative? This is the first time we are exposed to the scope of the relevance, is it classes all along? I was expecting a similarity score (in [0,1]). This should be made clearer. Since each baseline highlights a different aspect, did you consider combining some of them together to make a good (apples-to-apples) baseline? I am not completely convinced that the results are actually significant. Most improvements seem marginal.

W3: Originally I was under the impression that Section 2.2 covers the related work. I find the addition at the end not that useful. If the reader has to know more about the different aspects, I highly suggest expanding Section 2.2 at the expanse of Section 5. The Instruction Tuning bit can be beneficial in the right context under the model section.


Other comments:

C1: Why isn't the code available?
C2: Did you consider 2/3-citations? Why and why not?

**Questions:**

See under "Review"

**Reviewer Confidence:**

3: The reviewer is confident but not certain that the evaluation is correct

**Scope:**

3: The work is somewhat relevant to the Web and to the track, and is of narrow interest to a sub-community

---

### Official Review · Reviewer_ZjDk · 2024-11-30

**Novelty:** 6
**Technical Quality:** 6

**Review:**

### Quality
- Rigorous methodology with comprehensive experimental validation and strong comparisons to state-of-the-art methods.

### Clarity
- Well-structured with effective use of visual aids but challenging for non-experts due to technical jargon.

### Originality
- Novel integration of semantic, topic, and citation factors using a chain-of-factors approach with instruction-tuned models.

## Overview
This paper aims to propose a unified model for paper-reviewer matching that jointly considers multiple factors—semantic relevance, topic similarity, and citation connections—to improve the accuracy and scalability of assigning reviewers to academic submissions. The study applied experiments on four datasets with one of these data newly contributed by the authors of the paper, while the rest from previous studies.

The paper is technically robust and offers a novel approach to a critical problem in academic workflows. Its strength lies in the methodological integration of multiple factors and the practical applicability of its results. However, its focus on academic contexts and the complexity of its approach might limit broader applicability and adoption.

I have highlighted several areas that I consider to be limitations:

-**Scope of Application**: The model is designed for academic paper-reviewer matching, which may not generalise well to other domains requiring expertise matching.

-**Dependence on High-Quality Data**: The model's performance is contingent on well-structured metadata, including detailed publication profiles, citation links, and topical information, which may not always be readily available.

-**Fairness and Diversity**: The paper does not discuss fairness or diversity in reviewer assignments, which are critical considerations in real-world conference and journal contexts.


### Pros

#### Unified Model
- Combines semantic, topic, and citation factors into a cohesive framework, offering a comprehensive approach to paper-reviewer matching.
- The chain-of-factors methodology introduces a step-by-step, coarse-to-fine strategy, improving matching precision.

#### Innovative Use of Instruction Tuning
- Utilises instruction-tuned language models to produce factor-specific embeddings, ensuring each factor's unique characteristics are preserved.

#### Strong Experimental Validation
- Demonstrates consistent outperformance over state-of-the-art methods across multiple datasets in diverse fields.
- The inclusion of a newly annotated dataset (KDD) strengthens the experimental scope and contribution.

#### Relevance to Real-World Applications
- Directly addresses the scalability issue in large academic conferences, making the approach practical for high-stakes environments.


### Cons

#### Limited Scope Beyond Academia
- Focused exclusively on academic paper-reviewer matching, which narrows its broader applicability to other Web-based matching systems or general online recommendation systems.

#### High Complexity
- The chain-of-factors approach and instruction-tuned model add computational overhead, potentially limiting scalability in extremely large datasets.

#### Reliance on Pre-Existing Data
- Depends on well-structured metadata such as citation networks, topic annotations, and high-quality reviewer publication profiles, which may not always be available.

#### Lack of HCI Considerations
- The paper focuses on algorithmic efficiency but lacks discussion on usability or integration into existing conference management systems.

#### Potential Overfitting to Academic Context
- The datasets used are tailored to academic conferences, which may limit generalisation to other types of reviewer-expert matching scenarios (e.g., industrial peer reviews or open platforms).

**Questions:**

- The chain-of-factors approach adds a coarse-to-fine filtering mechanism. How does the computational cost of your method scale with extremely large datasets or conferences, and are there optimisations that can reduce this overhead without sacrificing accuracy?

- Your model relies on structured metadata like citation networks and publication profiles. How well do you think your approach would generalise to less structured domains, such as expert matching in online communities or industrial review processes?

**Reviewer Confidence:**

1: The reviewer's evaluation is an educated guess

**Scope:**

4: The work is relevant to the Web and to the track, and is of broad interest to the community

---

### Official Review · Reviewer_CxBr · 2024-12-01

**Novelty:** 4
**Technical Quality:** 5

**Review:**

Summary:

In the context of paper reviewing, this paper proposes a new approach called Chain-of-Factors for matching paper submissions to reviewers with three distinct factors. Contrary to previous approaches the authors consider semantic, topic, and reference factors all together in a multi-step process by training two text encoders that encode the instruction and the considered papers. TCOT outperforms SOTA paper-reviewer matching methods over multiple benchmarks and significantly improves over previous methods.

Summary Of Strengths:

* The paper makes a valuable contribution to the field by presenting a detailed description of the Chain-of-Factors approach, demonstrating potential applicability to a broader range of retrieval and ranking problems.
* Additionally, introducing a new dataset is a significant resource for the research community, offering opportunities for future work in this area.
* Overall, the paper is a strong contribution with room for refinement in specific areas.

Summary Of Weaknesses:

* The paper would have benefited from the evaluation of a larger model architecture. Reliance on a BERT model feels somewhat outdated, given recent advancements.
* The explanation of the KDD dataset is too brief, lacking an overview of its distribution, which would have provided deeper insights into its composition and utility.

**Questions:**

In line 298, you explain S twice (the first time in line 285). It seems you wanted to explain T instead.

It would have been interesting to discuss concrete use cases, such as through a field study conducted at a real conference. This would strengthen the practical relevance and demonstrate the real-world applicability of the proposed approach.

**Ethics Review Description:**

–

**Reviewer Confidence:**

3: The reviewer is confident but not certain that the evaluation is correct

**Scope:**

3: The work is somewhat relevant to the Web and to the track, and is of narrow interest to a sub-community

---

### Official Review · Reviewer_NQtN · 2024-12-03

**Novelty:** 6
**Technical Quality:** 5

**Review:**

The paper presents a model that considers three factors, one after the other, to filter out and score the fit of a reviewer with a paper under review. Relevancy with respect to semantics, topics covered, and authoring cited articles is examined, with a chain-of-thought type of process. Additionally, the authors contribute by adding a dataset of KDD-reviewed papers.

The problem is very well-defined and motivated. The related work is presented in a meaningful way, demonstrating also the understanding of the domain. The results support the statements made. and evaluate different parts of the proposed model. The appendix was also very useful.

However, even though most of the paper is well-written, sections 3.2-3.4 are not well-placed in the paper. The bigger picture is not presented, but rather, the manuscript focuses on smaller parts of the model without combining them all together.

**Questions:**

Is the code and the data (KDD dataset) publicly available?

**Reviewer Confidence:**

3: The reviewer is confident but not certain that the evaluation is correct

**Scope:**

3: The work is somewhat relevant to the Web and to the track, and is of narrow interest to a sub-community

---

### Official Review · Reviewer_edrz · 2024-12-10

**Novelty:** 5
**Technical Quality:** 5

**Review:**

This work tackles a very practical, interesting, and unique problem of paper-reviewer matching. They propose the new chain-of-factor model, a unified model for paper-reviewer matching that jointly considers semantic, topic, and citation factors.

This work seems to be very practical and relevant to the web application (e.g., openreview).

At the same time, they formulate their problem mathematically well throughout the paper, in addition to experimental results.

They propose a newly annotated dataset (KDD).

Also, they conducted extensive experiments with prior benchmark datasets with SOTA methods.

**Questions:**

It would be more helpful if authors can provide more details on how a new KDD dataset is annotated and curated in detail. It is in Appendix. However, some details can be present in the main section of the paper.

Why there are some missing values in Table 2? Would be great, if authors can explicitly provide the explanation.
Can you explain why two-sided Z test would be the right statistical test ? I think Z test assumes the normal distribution. Have you separate check/test the normality?


Newly created dataset be releasable ? I think it would be valuable for the community,  if authors can provide.

Contribution paragraph/section can be more concise in writing.

I still believe this work is very useful. However, dataset collected are somewhat old (2020 or earlier), which does not reflect the current skyrocketing paper submissions.

Would be great if authors can test with the more recent conferences for future work.

**Reviewer Confidence:**

2: The reviewer is willing to defend the evaluation, but it is likely that the reviewer did not understand parts of the paper

**Scope:**

4: The work is relevant to the Web and to the track, and is of broad interest to the community